# Morphological variation associated with trophic niche expansion within a lake population of a benthic fish

**Chiharu Endo❶\*, Katsutoshi Watanabe\***

Laboratory of Animal Ecology, Graduate School of Science, Kyoto University, Kitashirakawa-oiwakecho, Kyoto, Japan

\* chiharupn1622@gmail.com (CE); watanak@terra.zool.kyoto-u.ac.jp (KW)

**Data Availability Statement:** All relevant data are within the manuscript and its Supporting Information files.

**Funding:** This work was partly supported by JSPS (Nos. 26291079, 26250044, 17H03720,

## Abstract

Ecological theory suggests that generalist species should have traits with multiple adaptive peaks. Consequently, in heterogeneous environments such adaptive landscapes may lead to phenotypic divergence that becomes fixed in populations via reproductive isolation, thus driving speciation. However, contrary to this expectation, the process of ecological diversification in wild populations is not always associated with obvious trait divergence and reproductive isolation due to some ecological and geographic constrains. To examine the ecological conditions that promote (or inhibit) divergence is quite important to improve our understanding of the underlying mechanisms. Here we examine how the patterns of trait variation (divergence/non-divergence) are determined in relation to ecological niche expansion and gene flow using a benthic fish, *Pseudogobio esocinus*, in the Lake Biwa system, Japan. The fish exhibited various patterns of morphological variation in mouth parts among populations. Lake fish tended to have a smaller mouth compared with river fish and also showed remarkable individual variations within some local samples. Lake fish utilized chironomid larvae as the primary prey, as in riverine fish. But, fish with smaller and narrower mouths utilized significantly higher proportions of amphipods (a novel prey unique to the lake) as their secondary prey. Microsatellite analysis detected no genetic structuring in the Lake Biwa catchment, suggesting no reproductive separation among eco-morphologically divergent individuals. Our results exemplify population niche expansion associated with continuous eco-morphological variation without divergence, and provide insights into the role of non-discrete diversification for thriving in heterogeneous environments.

## Introduction

Trait variability leading to ecological niche expansion is an important factor contributing to intra- and interspecific diversity [1–3]. In natural systems, species often exhibit remarkable phenotypic variation across their geographic ranges [4], where much of the variation is explained by adaptation to the environments and resources in their habitat [1,5,6]. The extent of intraspecific phenotypic variations can vary among species; notably, habitat generalists that

18H01330) and Kyoto University Core Stage Backup Research Grant. The funders had no role in study design, data collection and analysis, decision to publish, or preparation of the manuscript.

**Competing interests:** The authors have declared that no competing interests exist.

occupy a wide range of habitats often exhibit higher variability by which they use more various niche conditions than habitat specialists [7–9].

Theoretically, when diverse habitats are available, there should be multiple adaptive peaks for traits that lead to discrete phenotypes under divergent natural selection [1]. If divergent selection coupled with assortative mating promotes reproductive isolation, the resulting poly-morphism may lead to ecological speciation without an extrinsic isolation barrier [10–12]. This kind of discrete polymorphism with ecologically important trait divergence has been doc-umented by a number of previous studies across animal taxa [10,13]. For example, freshwater fish often exhibit dimorphism in which two specialist morphs show the divergence of a feeding trait, reflecting adaptation to different niches, such as planktivorous morph in pelagic habitat and benthivorous morph in littoral habitat (i.e., resource polymorphism) [10,14]. However, the process of ecological diversification is not always involved with obvious trait divergence and resulting reproductive isolation. Traits under selection may exhibit non-discrete unimodal patterns (e.g., stickleback) [15] depending on some ecological, genetic, or developmental con-strains [16]. Under such constrains, trait divergence may be inhibited by unrestricted gene flow. However, it is not well understood how ecological and genetic factors affect the diver-gence patterns of traits and what conditions prevent population divergence [12,17,18].

*Pseudogobio esocinus*, a benthic cyprinid fish commonly found in Japan, exhibits large phe-notypic variation in a variety of morphological traits [19]. We have noticed that the species especially in Lake Biwa, the largest lake in Japan, exhibits highly diverse mouthpart character-istics, though the ecological and genetic background has been unclear. The species is a typical bottom dwelling fish that forages on benthic invertebrates buried in the sandy bottom, and it hides itself in the bottom sand when sensing danger. The mouthpart morphology is specialized to such a benthic lifestyle, and thus its characteristics should be quite important for survival reflected by efficiency of feeding and risk avoidance [19,20]. The pattern of variation is expected to be linked with bottom environments (see Materials & methods) and especially prey resources available in the environments [21,22]. Thus, this fish species in the Lake Biwa system is a good example in which to examine how the variation in ecologically important traits correlates with niche uses, and why large variability is maintained within the species and populations.

In this study, we aimed to clarify the patterns and causes of eco-morphological diversifica-tion among and within local populations of *P. esocinus* in the Lake Biwa system. To achieve this, we first investigated the characteristics and spatial patterns of morphological variation using specimens collected from the different benthic habitats in Lake Biwa and the surround-ing rivers. We then tested the relationship between morphological variation and niche diversi-fication by investigating the diet of *P. esocinus* living in contrasting bottom environments. We also estimated genetic population structure of this species in the whole Lake Biwa system to examine possible population subdivision and positive assortative mating relating to morpho-logical variation. Based on these results, we document a case of niche expansion with non-dis-crete morphological variation that is maintained within and among populations. We discuss the maintenance mechanisms of ecologically important trait variation in relation to success in persisting in heterogeneous environments.

## Materials & methods

### Ethics statement

This study was performed in accordance with the Fisheries Act in Japan and was conducted under permission for fish sampling in Lake Biwa from the local government (Shiga Prefec-ture). No ethical permission is required for described scientific sampling with fixed nets and

cast nets according to the Shiga Prefecture Fisheries Adjustment Regulations. The described methods were carried out complying with the Regulation on Animal Experimentation at Kyoto University. Accordingly, no ethical permission is required for described scientific activities and all experimental protocols were approved by the Kyoto University Animal Experimentation Committee.

## Study area

Lake Biwa is a representative ancient lake in East Asia over 400,000 years old and the largest lake in Japan (surface area 670 km$^2$, mean depth 41 m, and maximum depth 104 m) [23,24]. A large, deep pelagic zone primarily characterizes the unique environment of Lake Biwa, where various types of bottom environments in the littoral area (i.e., sandy, pebbly, and rocky bottoms) also provide diverse habitats for lake inhabitants [21] (Fig 1). The lake harbors more than 60 freshwater fish species, including a dozen endemic species (or subspecies) that have evolved unique lifestyles adapted to representative habitat types [23,24,25]. For example, several endemic species (e.g., the gudgeon *Sarcocheilichthys biwaensis* and the catfish *Silurus lithophilus*) and unique ecomorphs (e.g., a long-head type of *Sarcocheilichthys variegatus microoculus*) are found only in the rocky bottoms of the lake. In contrast, *Pseudogobio esocinus*, a typical generalist species, distributed widely in Japan except for Hokkaido and Ryukyu islands, is found not only in various types of bottom environments in the lake, but also in the rivers surrounding the lake.

## Fish sampling

The specimens of *P. esocinus* were collected mainly by using a cast net or fixed net from nine sites in Lake Biwa (L1–L9) and nine sites in different rivers surrounding the lake (R1–R9) between 1993 and 2015 (mostly between 2007–2015; Fig 1, Table 1). Following capture, all fishes were immediately removed from nets and euthanized on ice. Then those specimens received injection of 10% formaldehyde solution into the body cavity to restrain digestion for diet analysis. Fin clips were preserved in 99% ethanol for DNA extraction, and whole-body samples were fixed in 10% formaldehyde for morphology and diet analyses. Our samples included specimens from the collection at the Lake Biwa Museum (Shiga Prefecture, Japan), the registration numbers: 1210016849, 1210017123, 1210017818, 1210018760, 1210023880, 1210027208, 1210028811, 1210029013, 1210031389, 1210031774, 1210031802, 1210031838, 1210032078 (used for morphological analysis).

## Morphological analyses and anatomical observations

We conducted observations and measurements of body shape, with special focus on mouth-part morphology [19]. For the shape analysis of body and mouth, we used a total of 389 specimens collected from six lake sites (L1–L6) and seven river sites (R1–R7) (Fig 1, Table 1), which represented samples from all different bottom environments, i.e., sandy, pebbly, and rocky bottoms [21]. To quantify the variation in shape, we measured a total of eight distances between landmark points on the body by vernier caliper to the nearest 0.1 mm (Fig 2). The measured traits were standard length (SL) as a proxy of body size, body depth (BD), body width (BW), caudal peduncle depth (CPD), head length (HL), snout length (SnL), mouth length (ML), and mouth width (MW) (Fig 2). We used only fish whose standard length was 60 mm or larger (Table 1) to avoid conspicuous effect of allometric changes with growth [19].

To identify the major morphological variation in the whole samples, we conducted principal component analysis (PCA) for the standardized values of the above eight measurements using function prcomp in the R software ver. 3.0.3 (R Core Team 2014). All the measurements were summarized into principal components (PCs). Because the first PC (PC1) was expected

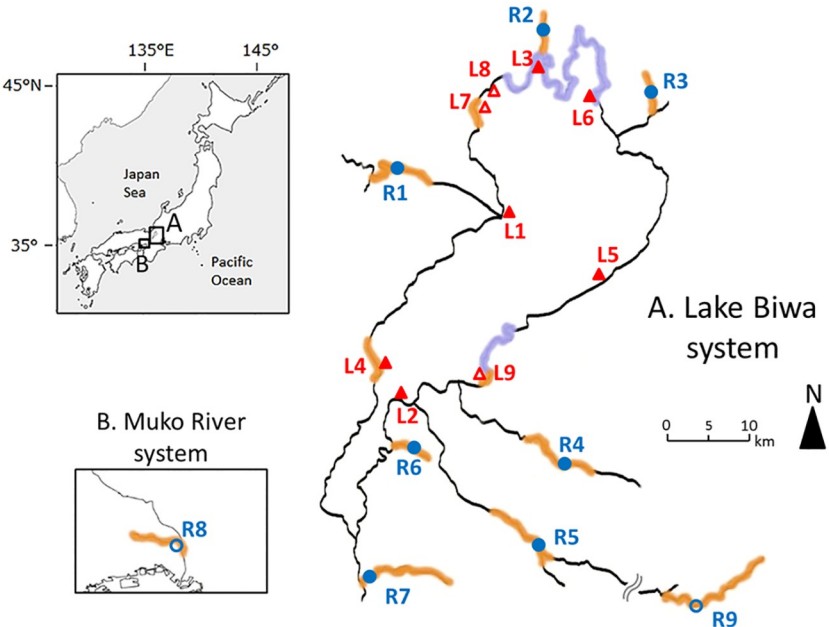

**Fig 1. Sampling locations around Lake Biwa.** Red triangles indicate lake sites (L1–L9) and blue circles indicate river sites (R1–R9). Samples captured in the sites with filled symbols were used for both morphological and molecular analyses, and those with open symbols were only for molecular analysis. These sampling sites are generally categorized as rocky zone (violet shadow), pebbly zone (orange shadow), and sandy zone (no shadow) based on the bottom environments.

**Table 1. Locality name, geographic coordinate, sampling year, and sample size for specimens used in this study.**

| Code | Locality | Sampling year | Morphology* | Microsatellite | Diet |
|------|----------|---------------|-------------|----------------|------|
| L1 | Adogawa (35.31N, 136.08E) | 2013, 2014 | 50 (118.4–192.3) | 26 | - |
| L2 | Moriyama (35.12N, 135.94E) | 2013 | 33 (81.6–155.8) | 28 | - |
| L3 | Oura (35.48N, 136.11E) | 2013 | 47 (112.3–182.0) | 28 | - |
| L4 | Wani (35.16N, 135.93E) | 2013–2015 | 60 (74.5–177.0) | 21 | 30 |
| L5 | Hikone (35.25N, 136.19E) | 2013 | 52 (115.7–180.0) | 26 | - |
| L6 | Onoe (35.45N, 136.18E) | 2007 | 21 (60.0–159.0) | 22 | 16 |
| L7 | Momose (35.44N, 136.06E) | 2007 | - | 20 | - |
| L8 | Kaizu (35.45N, 136.07E) | 2007 | - | 12 | - |
| L9 | Omihachiman (35.15N, 136.05E) | 2007 | - | 20 | - |
| R1 | Adogawa R. (35.35N, 135.92E) | 1999**, 2015 | 21 (100.5–151.0) | 20 | 13 |
| R2 | Oura R. (35.49N, 136.12E) | 2015 | 11 (60.0–148.2) | 21 | - |
| R3 | Tagawa R. (35.45N, 136.26E) | 2015 | 14 (62.5–121.8) | 16 | - |
| R4 | Hino R. (35.06N, 136.17E) | 1999–2001**, 2007, 2015 | 19 (68.0–134.5) | 18 | - |
| R5 | Yasu R. (34.96N, 136.13E) | 2001**, 2007, 2014, 2015 | 10 (67.1–127.9) | 10 | - |
| R6 | Moriyama R. (35.06N, 135.97E) | 2007, 2015 | 30 (61.8–129.0) | 26 | - |
| R7 | Daito R. (34.94N, 135.92E) | 1993–2001**, 2007 | 21 (64.3–131.8) | 11 | 9 |
| R8 | Muko R. (34.77N, 135.37E) | 2013 | - | 10 | - |
| R9 | Tamura R. (34.92N, 136.31E) | 2007 | - | 10 | - |

Code: L, locality in Lake Biwa; R, locality in rivers around the lake.

*The ranges of standard length of specimens are indicated in the parentheses.

**Specimens kept in Lake Biwa Museum.

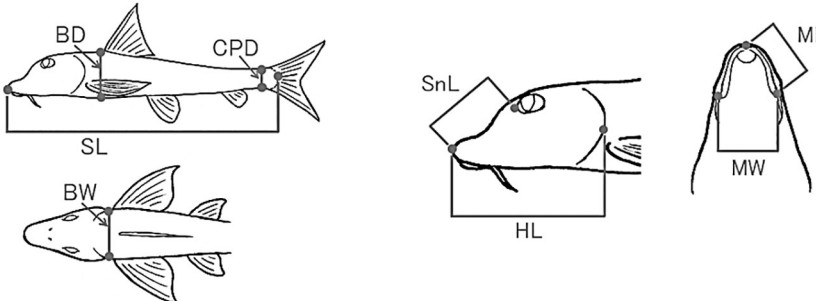

**Fig 2. Measurement parts of morphological traits for *Pseudogobio esocinus*.** Standard length (SL), body depth (BD), body width (BW), caudal peduncle depth (CPD), head length (HL), snout length (SnL), mouth length (ML), and mouth width (MW).

to account for body size variation and showed very high correlation with SL ($r^2 = 0.934$), we treated the rest of components (PC2, PC3, etc.) as shape variables independent of body size. We calculated the contribution ratio of each principal component in the variation excluding PC1's contribution. The first two components (PC2 and PC3) accounted for nearly 80 percent of the variation independent of body size variation (S1 Table). Therefore, we considered these two principal components as the effective shape factors. The PC2 and PC3 represented mainly relative mouth size and mouth width, respectively (S1 Table). We conducted Silverman's tests to check the modalities (unimodal or multimodal) of those component scores for the samples in Lake Biwa, rivers, and each locality, using function modetest in the R package "multimode" [26].

To reveal the patterns of morphological variations across Lake Biwa and rivers, we conducted several sets of comparisons using lake and river specimens. First, to test for differences in the distribution of PC2 and PC3, we performed a generalized linear mixed model (GLMM) analysis using function glmer in R. The model with Gaussian distribution was integrated by PC scores as the objective variable with the category of lake/river (fixed effect) and the site codes (random effect) as the explanatory variables. To test for differences in the degree of overlapping/differentiation in morphological characteristics among sites, we conducted analyses of variance (one-way anova) for the two PC scores of lake samples and river samples independently. We then compared the components of F-values between lake and river samples to determine how much each of variations within sites and among sites contributes to the total variation in each habitat. Further, to quantify the degree of morphological variation within a sample at a site, we defined the degree of variability, "PV*i*", as the phenotypic deviation of individual specimens from the average value of the local sample. This was calculated as the square of multidimensional Euclidean distance from the average values of PC2 and PC3 in Eq (1):

$$\text{PV}i = \left(x_{pc2,i} - \overline{x_{pc2}}\right)^2 + \left(x_{pc3,i} - \overline{x_{pc3}}\right)^2, \tag{1}$$

where $x_{pc2,i}$ and $x_{pc3,i}$ are the second and third principal component scores of the individual $i$ in a sample, respectively. Next, we tested for differences in PV*i* between lake specimens and river specimens using GLMM with gamma distribution with the same explanatory variables as in the above comparisons of PC scores.

To link the variation of mouthpart measurements with kinematic mechanisms of the mouth movements, we studied the anatomical structure of mouth parts for some typical *P. eso-cinus* specimens selected among various mouth types (large-wide to small-narrow). Making

clear and double-staining specimens following Kawamura and Hosoya's protocols [27], we observed the movements of bony elements along with the mouth being opened and closed.

## Diet analyses

Based on the morphological analyses, we selected four local samples that were representative of the patterns of mouthpart variation (n = 9–30 for each sample; same as specimens use in morphological comparison, excluding those without gut contents). Because the lake specimens tended to show large, overlapping variation among sites, we selected the two sites extreme in variability (L4 with large variation at a pebbly site and L6 with small variation at a rocky–pebbly site; Fig 1). The river samples tended to show less overlapping, discrete variation especially in PC3 (mouth narrowness); thus, we selected two river samples that showed distinctive characteristics (R1 and R7).

The gut contents of each specimen were identified to the lowest taxonomic level as possible and categorized into ten groups under a stereo microscope: chironomids, mayflies, caddisflies, blepharicerids (Insecta), amphipods, copepods, ostracods, cladocerans (Crustacea), oligochaetes (Oligochaeta), and hydrachnids (Hydracarina). We evaluated the relative contribution of each food item to the contents of individual fish using the points method, which gives scores to each category based on the proportion by approximate volume [28]. The scores of each prey category were allotted by counting the number of cross-points of 1-mm spaced grid covered with the gut contents spread over a plate and summing up the total points for the category.

To examine the effect of mouth shape on the proportion of each food item, we conducted GLM multiple regression analysis in the R software. The models were composed of total points for every food item with enough data to properly calculate statistics as the objective variable, and PC2, PC3, SL, and total points of all food items (total P) as the explanatory variables. We first fitted GLM models with the Poisson distribution, but they indicated over-dispersion of our count data in the Poisson model. We therefore use a quasi-Poisson model as an alternative solution for data sets typically exhibiting over-dispersion. The data sets of amphipods and oligochaetes in L4, however, contained many zero-values (amphipods, 50%; oligochaetes, 50%), and we then conducted a zero-inflated Poisson (ZIP) model using the package pscl [29], which was originally designed to model empirical count data sets exhibiting over-dispersion and/or excess zeros [29,30].

## Molecular analysis

*Pseudogobio esocinus* in western Japan includes two largely differentiated mitochondrial DNA groups (Group A and Group B) [31], corresponding to two cryptic species [32]. All samples from the lake (L1–L9) and rivers (R1–R8) were of Group A, except a part from R9 that possessed the Group B mtDNA. To examine the potential population subdivision in *P. esocinus* in and around Lake Biwa, we conducted microsatellite analysis for a total of 345 specimens from nine lake sites (L1–L9) and nine river sites (R1–R9) (Fig 1, Table 1). The sampling sites included almost all types of bottom environment where *P. esocinus* inhabits (Fig 1, Table 1). We screened 48 microsatellite loci developed by Takeshima et al. (2016) [33] using specimens from the mtDNA group A and B and developed 14 microsatellite primer pairs available for the both groups (S2 Table).

We extracted DNA from fin clips using a Genomic DNA Purification Kit (Promega, Madison, Wisconsin, USA) and performed PCR amplification in a 10 μl volume, containing 3.8 μl ultrapure water, 5 μl Type-it Microsatellite PCR Master Mix (Qiagen), 0.2 μl of each 1 μM primer, and 1 μl of DNA template. The PCR settings consisted of the first step (denature, 95˚C, 5 min), 35 cycles of the second step (denature, 94˚C, 15 s; annealing, 58˚C, 30 s;

extension, 72°C, 30 s), and the last step (extension, 60°C, 30 min). We sized PCR products on an automated DNA sequencer (ABI 3130xl, Applied Biosystems, Foster City, CA, USA) with HiDi and GeneScan 500 LIZ dye size standard (Applied Biosystems), and scored allele sizes using the software GeneMapper (Applied Biosystems).

We selected fourteen microsatellite primer sets for *Pseudogobio esocinus* (S2 Table) after checking the presence of null alleles using MICRO-CHECKER [34] and tested for deviation from Hardy-Weinberg equilibrium (HWE) at each locus using ARLEQUIN ver. 3.5.2 [35]. We found three loci displaying evidences for null alleles (Pes1_14, Pes1_21, Pes2_05; 5% significance level) and one to six loci in each sample displaying evidence of deviation from HWE (5% significance level). However, we included all those loci for data analyses since they might be caused by some ecological factors, e.g., presumable hybridization between two mtDNA groups in R9, or immigrations of river individuals to Lake Biwa.

To examine the degree of gene flow and genetic differentiation among local samples, we estimated population structure using a Bayesian clustering approach in the software STRUCTURE ver. 2.3.4 [36]. Analyses presuming the number of clusters (referred by K) from 1 to 6 were performed with 10 replicates for each K value, with a burn-in period of 50,000 steps followed by 100,000 Markov chain Monte Carlo (MCMC) iterations under the admixture model and assumption of correlated allele frequencies among populations. We calculated ΔK (rate of change for log likelihood respect to K) to determine the best estimation of K, following Evanno et al. (2005) [37].

We estimated the pairwise-Fst between local samples by the ARLEQUIN and tested the significance of population differentiation under Holm's correction (significance level = 0.05). We also calculated allelic richness (Ar) using FSTAT ver. 2.9.3 [38] and mean observed (Ho) and expected (He) heterozygosities using ARLEQUIN as the indices of genetic diversity. We then examined differences in these indices of genetic diversity between lake and river samples using a Mann-Whitney U test.

To grasp the presence/absence of assortative mating relating to individual morphological variations within Lake Biwa, we tested the correlation between pairwise genetic distance and phenotypic distance of individuals in all the lake specimens for which their morphology was analyzed (L1–L6). We calculated the matrix of genetic distance based on microsatellite data with a delta mu 2 index using the software Population ver. 1.2.31 [39], and that of phenotypic distances defined by the absolute values of differences between PC scores of individual specimens for each of PC2 and PC3 and also by the Euclidean distance (ED) for PC2 and PC3. Significance of the correlation between the genetic distance matrix to the phenotypic distance matrix was tested by a Mantel test with 999 permutations in the R package vegan [40]. We also evaluated genetic distance as the difference of genomic composition, calculated as the absolute difference of proportion of one of clusters (q1 scores) estimated by STRUCTURE analysis (q-value distance). We examined the significance of the correlation between the q-value distance matrix and phenotypic distance matrix for each of PC2, PC3, and ED with a Mantel test.

## Results

### Patterns of morphological variation

Excluding the influence of body size variation (PC1), morphological variation among and within local populations of *P. esocinus* in Lake Biwa and surrounding rivers were mostly explained by variations in mouth parts. PC2 mainly represented variation of mouth size (positive contribution of ML and MW) and PC3 represented mouth narrowness (positive and negative contribution of ML and MW, respectively) (S1 Table). The PC2 and PC3 exhibited unimodal distributions for pooled samples of the lake or rivers (Silverman's tests, p ≥ 0.57, Fig 3) and for all respective local samples (p ≥ 0.33; S1 and S2 Figs).

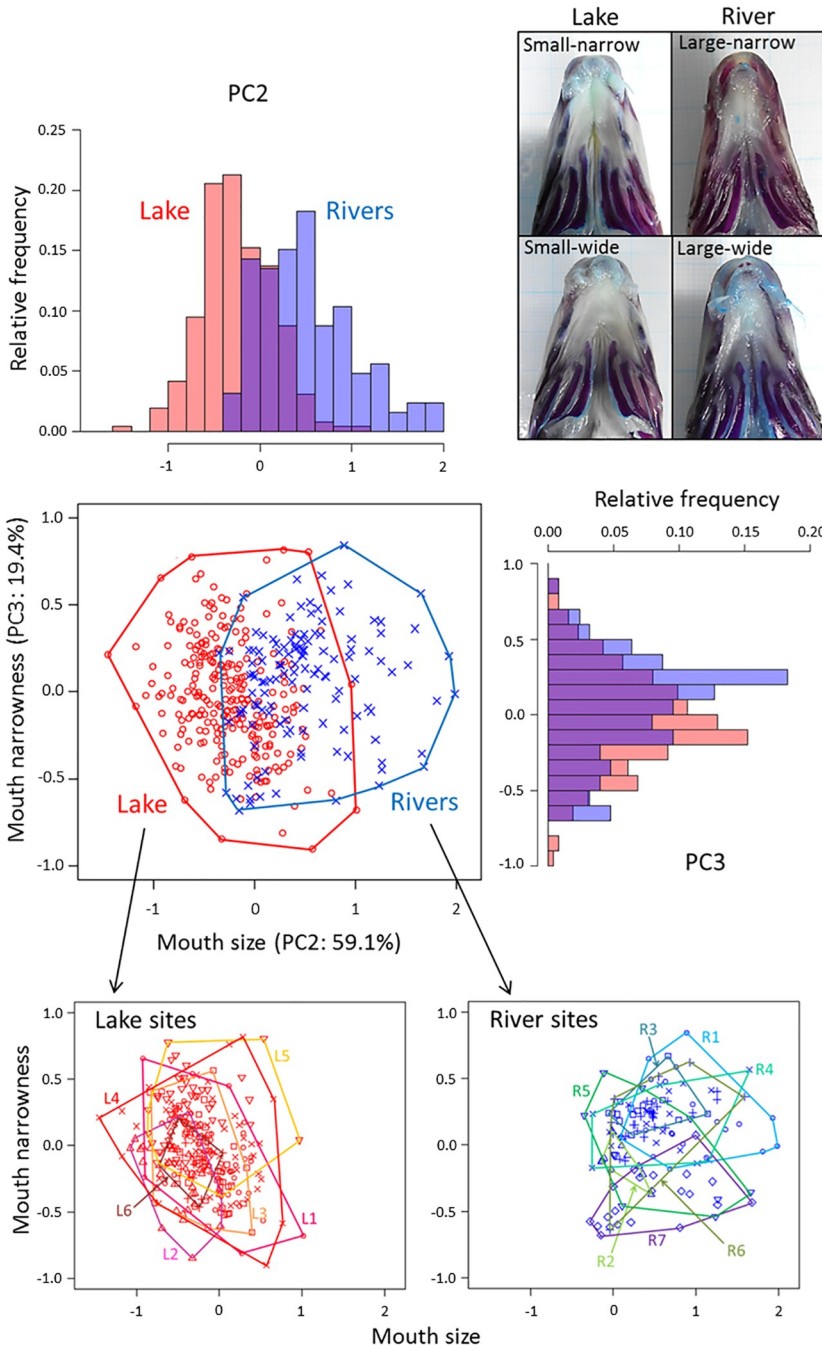

**Fig 3. Scatter plots and histograms of PC2 and PC3.** Contribution ratio of each principal component in the variation excluding PC1's contribution is shown in parentheses. Specimens of *Pseudogobio esocinus* were collected from Lake Biwa and the surrounding rivers (red, lake sites; blue, river sites). Distributions of PC2 and PC3 scores showed unimodality both in lake samples (Silverman's tests, PC2, p = 0.57; PC3, p = 0.74) and river samples (PC2, p = 0.71; PC3, p = 0.77). The bottom panels are scatter plots of PC2 and PC3 for each local sample in the lake (left) and rivers (right). Photos are the ventral views of representative specimens with wide and narrow mouths of the specimens collected from Lake Biwa (narrow, L4; wide, L3) and rivers (narrow, R1; wide, R6).

Lake specimens tended to have a smaller mouth size (PC2) compared to river specimens (GLMM, t = 6.63, p < 0.001, 3 and S3 Fig). The extent of mouth size variation among local samples was not clearly different between Lake Biwa and rivers (F-value in one-way anova, lake samples, F = 9.67; river samples, F = 7.43, 3 and S3 Fig). In mouth narrowness (PC3), there were no significant differences between lake and river specimens (GLMM, t = 0.798, p = 0.39). However, the F-value in anova of lake samples (F = 18.9) was smaller than that of river samples (F = 31.4), indicating a larger morphological overlap among the lake samples (3 and S3 Fig).

Local samples both of Lake Biwa and rivers showed various extents of variations in the integrated phenotypic characteristics of mouth size and narrowness (PV*i*; S3 Fig). Especially, high phenotypic variability were found in L4 and R5, and low variability were found in L6 and R2. No significant differences in PV*i* were found between lake and river samples (GLMM, t = -1.02, p = 0.335).

The dimensional differences in mouth size and narrowness were linked with the mechanistic difference in movements of mouth parts, as showing in Fig 4. The lake specimen with a wide mouth exhibited a markedly downward mouth opening (Fig 4A). The mouth movements in the river specimens, which tended to have larger and wider mouths than lake specimens, were similar to this. On the other hand, the lake specimens with a narrow mouth showed more forward mouth opening (Fig 4A). The different degrees and directions of mouth opening were attributed to the related bony structure, i.e., the different motion ranges of a small median bone (kinethmoid) to shift upper jaw bones (Fig 4B). These kinematic differences were not discrete but continuous, various mouth movements being shown depending on mouth shapes.

## The relationship between morphology and diet

Diet analysis for *P. esocinus* living in contrasting benthic habitats at four sites indicated that their dominant prey item was chironomid larvae in both lake and rivers (62.4–98.6% in volume of all prey items; Fig 5). The secondary prey item for lake samples was amphipods (L4, 15.0%; L6, 34.1%), whereas that for river samples was mayfly (R1, 8.2%; R7, 1.0%) or caddisfly larvae (R1, 7.3%; R7, 0.3%).

There was significant association between morphological characteristics and diet in the L4 sample, which showed large variation in mouthpart morphology (Fig 6, S3 Table). In this sample, fish individuals contained amphipods in their diet with various proportions (0–54.3% with 11.9% on average; S4 Fig). Mouth size (PC2) had a significantly negative effect on the usage of amphipods (GLM with ZIP model, z = -4.53, p < 0.001), and mouth narrowness (PC3) had a significantly positive effect on that (z = 4.54, p < 0.001) (S3 Table). We also found significant effects of mouthpart morphology on the usage of oligochaetes in L4 (PC2, z = 2.09, p = 0.037; PC3, z = 2.27, p = 0.023); however, the proportion of this item was more explained by the effect of body size (SL, z = 4.79, p < 0.001) (S3 Table).

In contrast to L4, the specimens from L6, which possessed a less variable, small mouth, showed a high proportion of amphipods in the diet (0–97.3% with 31.4% on average; S4 Fig). Almost all individuals used amphipods to some extent, but the degree showed no significant correlation with the characteristics of mouthparts (GLM with quasi-Poisson model, PC2, t = 0.87, p = 0.40; PC3, t = 0.14, p = 0.89; S3 Table).

River specimens in R1, which were characterized as having a large, narrow mouth, included some amount of mayflies (7.3%) and caddisflies (0.3%) in addition to chironomids (83.9%). In this sample, the proportion of mayflies was correlated with body size (GLM with quasi-Poisson

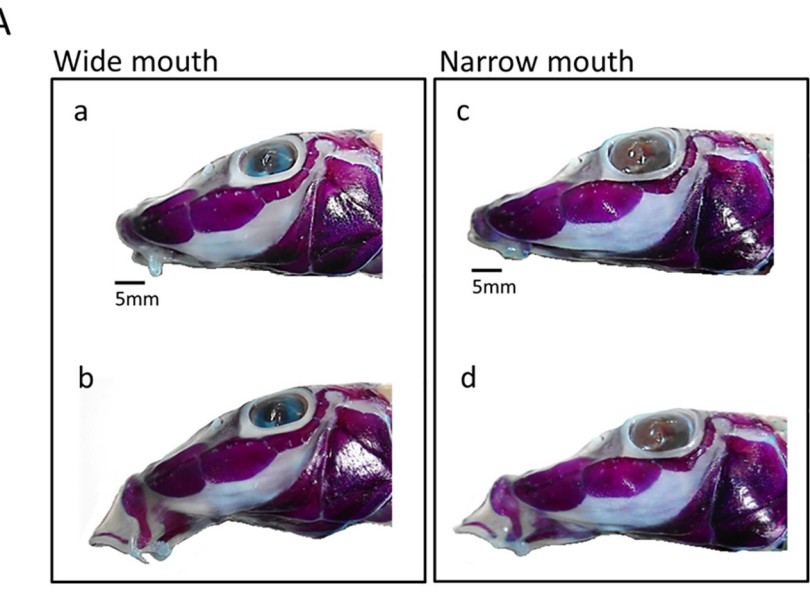

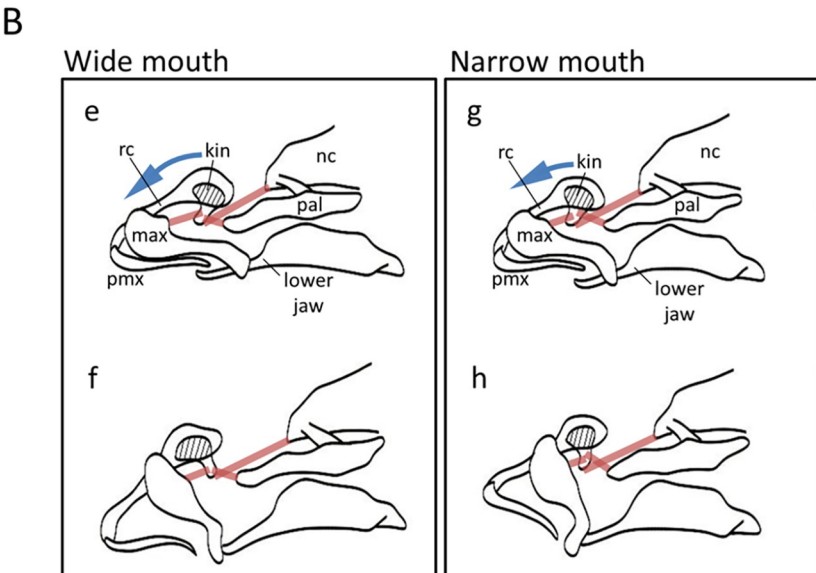

**Fig 4. Lateral views of head of the representative specimens in *Pseudogobio esocinus* collected from Lake Biwa.** Left, wide mouth; right, narrow mouth. (A) Double-staining specimens, (B) schematic bony structure and associated ligaments (pale brown lines) with showing kinematics of upper jaw protrusion. The retracted (a, c, e, g) and protruded states (b, d, f, h) of the mouths are shown. The kinethmoid (kin), a median sesamoid bone, connects the upper jaw bones with the neurocranium mediated by the ligaments and rostral cartilage (rc). The kinethmoid leans a little to the neurocranium when the jaws are closed, whereas it rotates rostrally during jaw protrusion. Blue arrows indicate different extents of rotation of the kinethmoid causing different degrees of upper jaw protrusion between wide and narrow mouth specimens. max, maxilla; pmx, premaxilla; nc, neurocranium; pal, palatine.

model, SL, t = 2.92, p = 0.02; S3 Table). On the other hand, river specimens in R7, which tended to have a large, wide mouth, showed the strongest specialization on chironomids (98.6%), and no significant correlations between individual morphology and the usage of this item (PC2, t = -1.94, p = 0.05; PC3, t = -0.58, p = 0.56; S3 Table).

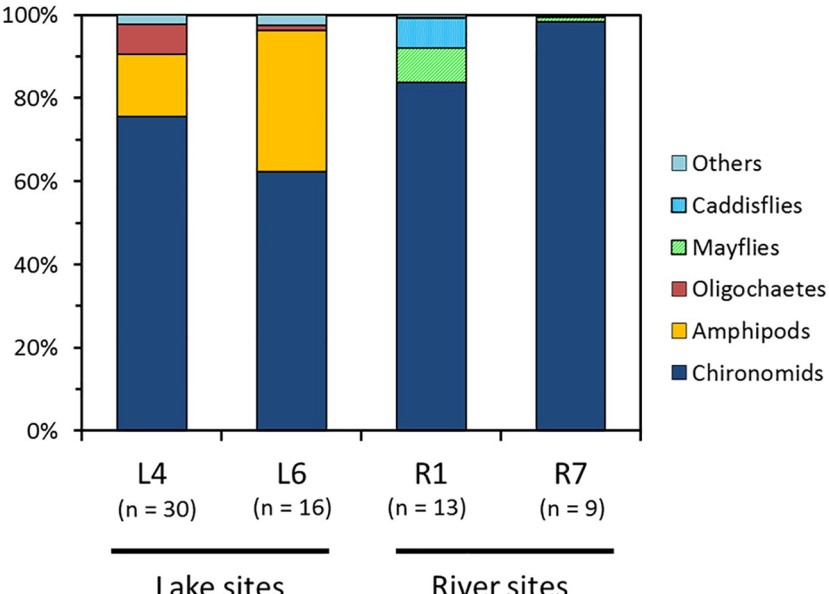

**Fig 5. Diet compositions of *Pseudogobio esocinus* in lake sites (L4, L6) and river sites (R1, R7).** The number of fish specimens used for diet analysis is shown in parentheses.

## Gene flow and population subdivision

Bayesian clustering analysis based on 14 microsatellite loci data showed very weak genetic population structuring in *P. esocinus* in Lake Biwa, although some degree of population structuring was detected in the whole Lake Biwa system. All the local samples in Lake Biwa and rivers (except R9 from a cryptic species [31,32]) shared two different genetic elements in various proportions (Fig 7).Their average proportion was similar among the local samples from Lake Biwa (about 0.7:0.3), whereas it was rather variable among the river samples. The pairwise-Fst indicated that no pairs between lake samples showed significant differentiation under Holm's correction (S4 Table). On the other hand, river samples tended to be differentiated from each other. Samples from the lake and rivers were generally differentiated, but some pairs did not show significant differentiation. Lake samples showed significantly higher values of allelic

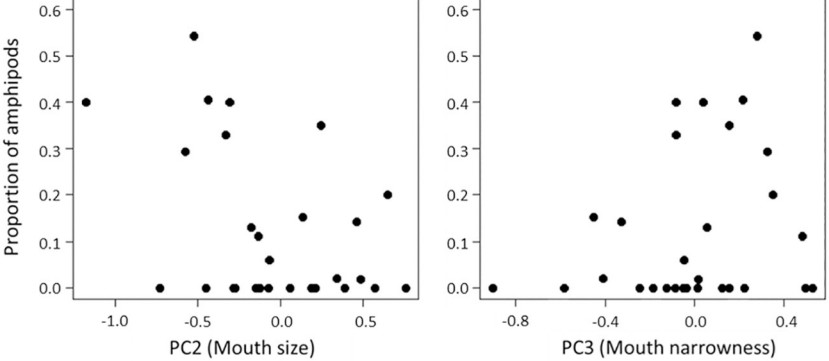

**Fig 6. Relationship between PC scores and the proportion of amphipods in diet of *Pseudogobio esocinus*.** All the fish specimens were collected from L4 in Lake Biwa.

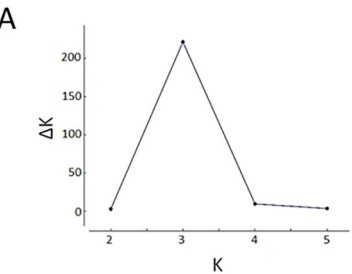

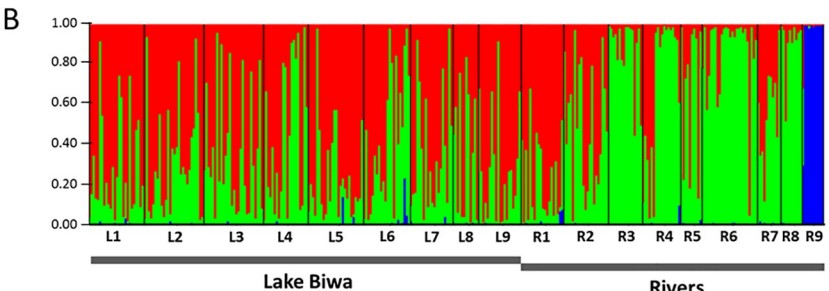

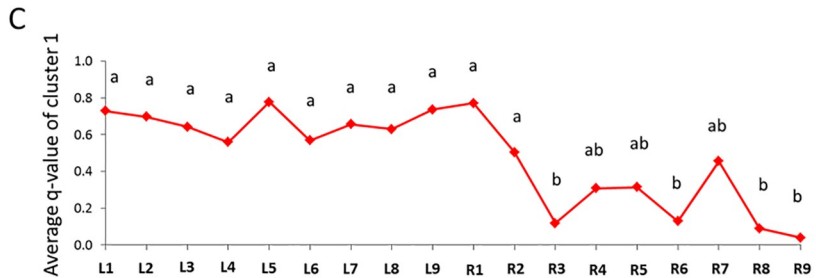

**Fig 7. Bayesian assignment analysis based on 14 microsatellite-locus data.** The samples of *Pseudogobio esocinus* were captured in Lake Biwa and the surrounding rivers. (A) ΔK as function of the number of assumed populations (K = 1–6). (B) Results of the assignment for K = 3. Each vertical bar indicates an individual partitioned into the three genetic clusters (cluster 1, red; cluster 2, green; cluster 3, blue). (C) The average q-value of cluster 1, which is the proportion of the genetic component in a location sample. Distinct letters indicate significant differences in that value in multiple comparison test under Bonferroni correction (significance level = 0.05).

richness than river samples (Mann-Whitney U test, p < 0.001; S5 Table), implying larger population size of the lake population comparing with river populations.

The genetic data did not support assortative mating relating to morphological characteristics in Lake Biwa specimens; there were no significant relationships between genetic distance and phenotypic distances for pairs of individuals (PC2, Mantel statistic r = -0.014, p = 0.65; PC3, Mantel statistic r = -0.031, p = 0.85; ED, Mantel statistic r = -0.035, p = 0.86, S5 Fig). Also, no significant relationship between the q-value distances and phenotypic distances for individuals was detected (PC2, Mantel statistic r = -0.012, p = 0.64; PC3, Mantel statistic r = -0.03, p = 0.92; ED, Mantel statistic r = -0.031, p = 0.88).

## Discussion

### Mouthpart diversification associated with niche expansion

The morphological variation of *Pseudogobio esocinus* in the Lake Biwa system was primarily explained by variation in mouth size and mouth width, excluding the effect of body size.

Although both lake and river specimens showed large variation in those characteristics, the former tended to have a smaller mouth than the latter. The small and narrow mouth is an extreme phenotype unique among specimens from Lake Biwa system, and involved the mechanistic, possibly functional changes in mouth movement in comparison with the common larger and wider mouth. This variation in mouth parts is suggested to reflect different feeding behavior and resource usage.

Indeed, morphological variation of *P. esocinus* was at least partly linked to the pattern of prey usage. In all the samples from various habitats, the primary prey item was chironomid larvae, which inhabit the sandy, pebbly, and even rocky bottoms commonly found in lakes and rivers [41]. This seems to be the general feeding habit in *P. esocinus* [19]. On the other hand, the secondary prey in the lake samples was amphipods, which rarely co-occur with *P. esocinus* in river habitat [42]. Amphipods in Lake Biwa (*Jesogammarus naritai*, *Jesogmmarus annandalei*, *Kamaka biwae*, and an alien species *Crangonyx floridanus*) usually live on the bottom surface or even in the water column with active movement [43,44]. So, for *P. esocinus* in Lake Biwa, amphipods are of a novel prey type, different from chironomids and other riverine benthos. At the single site in the lake (L4), the fish with a smaller and narrower mouth preyed on amphipods more frequently (Fig 6). Also, specimens from L6 commonly had a small mouth and used amphipods in a high proportion. From these correlations, the modification of mouthpart morphology in lake fish had some functional importance to expand their prey types, although the driving force (e.g., competition) to produce such prey differentiation is unclear.

There should be a trade-off in feeding effectiveness between the basic (large-wide) and novel (small-narrow) mouth types. *Pseudogobio esocinus* generally forages via suction of benthic substrates and subsequent sorting of prey from fine inorganic particles. Benthic suction is achieved by a downward protrusion of the jaw whilst opening and closing the mouth [19]. Thus, the larger and wider mouth with greatly protruding jaw, as observed most typically in the river sample at R7, would have an advantage in such suction feeding because of its larger buccal cavity that improves suction force [45–47]. It may be highly adaptive to feeding on the prey buried in the bottom in the middle reaches of rivers like R7. In contrast, the smaller buccal cavity of the small-narrow mouth type found in Lake Biwa may function less for suction feeding. Instead of improving suction force, however, the small-narrow mouth may have an advantage in picking up a moving prey by more forward mouth protrusion. To prove the differences in the feeding mode and efficiency between the mouth types, an experimental study is necessary.

## Within- and among-population variation and its maintenance mechanisms

The pattern of mouthpart variation contrasted between the river and lake populations. The river samples showed less overlaps in mouth width among sites. They were genetically isolated to various degrees, suggesting limited migration and gene flow among the rivers. This suggests that the divergence in mouth width among rivers reflects the local adaptation to some specific environmental condition (e.g., grain size of the substrate) at each habitat. In contrast, the local samples in Lake Biwa showed larger overlap in mouth width among sites. Also, the within-sample variation exhibited non-discrete, unimodal patterns. No genetic subdivision was observed for the whole lake population, and some gene flow existed even between lake and river populations. These observations imply that the variation in their mouth width does not simply reflect local adaptation. The variation of mouthpart morphology is supposed to be maintained under a meta-population structure

with various degrees of gene flow among local populations in whole Lake Biwa and the surrounding rivers.

Although local populations in the lake showed large overlap in mouthpart morphology with each other, their variation patterns exhibited remarkable differences. Some (typically L4) included various individuals with a small-narrow to wide mouth within a single site, but others consisted of only similar individuals with a small mouth (typically L6). These morphological patterns in the local populations were partly reflected in their diet, i.e., the proportion of different prey items (chironomids vs. amphipods). In Lake Biwa, the occurrence and abundance of benthos, including chironomids and amphipods, are known to vary among littoral sites mainly associated with substrate types [22,41,43], although we do not have quantitative data on the prey in our sampling sites. The patterns and extents of mouthpart variation (and diets) in local populations could be influenced by spatial patterns of prey availability.

Under high gene flow among local populations, how has their morphological variation within and among populations persisted? One possible reason that the variation has persisted could be explained by the selection–gene flow balance [12]. Since chironomid larvae were the most important prey item in all localities both in the rivers and lake, the ubiquitous prey item may weaken the selection for mouthpart morphology and hence alleviate the migration load for immigrant individuals. Also, the migration is not severely limited even between the rivers and lake since sandy or pebbly bottom habitat are distributed continuously in the rivers and the littoral zone of the lake, except for a few rocky areas [21,22]. Although fish with different morphological features may still restrictedly use a specific microhabitat with different availability of food resources, no assortative mating associated with morphology was detected. Unlike nuptial color [48–50], mouthpart morphology may be difficult to be used as a mate-choice signal. Thus, the limitations on the genetic/sexual isolation, as well as the indiscrete trophic pattern, could contribute to prevent the population divergence in this species.

Another possible explanation for the persistence of variation in functional traits is phenotypic plasticity [10,51,52]. The mouthpart morphology could be determined according to the ecological (prey) conditions during ontogenetic development. In such case, a panmictic population in the lake could produce various patterns of mouthpart variation within and among local populations reflecting local resource patterns. Since there is no information on heritability of the mouthpart morphology of this fish, crossbreeding and common garden experiments are necessary to determine the mechanisms that maintain trait variation within and among local populations.

Regardless of genetic control or phenotypic plasticity, the variability of mouthpart morphology would have been beneficial for *P. esocinus* in their colonization of the lake environment largely different from river habitat. The environmental characteristics of the present Lake Biwa (i.e., pelagic, deep, and rocky-shore areas) started to form 400,000 years ago, and harbors more than a dozen endemic fish species (e.g., cypriniforms, siluriforms, gobiiforms, and others) that have derived from riverine ancestors [25,53]. Pelagic vs. river/littoral, and deep vs. shallow environments are both contrasting environments, which often cause strong divergent selection in ecological and physiological traits in fishes (e.g., stickleback, Arctic charr, pumpkinseed sunfish etc.) [13,54,55]. Differentiation in reproductive sites (or seasons) involving with colonization of novel environments may have facilitated ecological speciation along those environmental differences [12,56,57]. In contrast, the sandy–pebbly bottom that *P. esocinus* uses is spatially heterogeneous but continuous environment in terms of grain size and composition of available food types (i.e., common chironomids and unique amphipods). Although *P. esocinus* is strongly constrained to live on the sandy–pebbly

substrate as a typical benthic species, it shows niche expansion presumably responding to local resource conditions without population divergence. This ability may be an important factor helping the species to increase in population size and thrive in both lake and riverine habitats.

## Supporting information

**S1 Table. Results of the principal component analysis (PCA) for standardized morphological trait values.**
(DOCX)

**S2 Table. Primer sequences and the related information of microsatellite primers for *Pseudogobio esocinus*.**
(DOCX)

**S3 Table. The results of GLM analyses on the proportion of each food item in gut contents of *Pseudogobio esocinus* in L4, L6, R1, and R7.**
(DOCX)

**S4 Table. Pairwise-Fst estimated from 14 microsatellite-locus data in Lake Biwa and the surrounding rivers.**
(DOCX)

**S5 Table. Summary of polymorphism of 14 microsatellite-locus of *Pseudogobio esocinus* in Lake Biwa and the surrounding rivers.**
(DOCX)

**S1 Fig. Histograms of PC2 and PC3 of local samples of *Pseudogobio esocinus* in Lake Biwa.** Distributions of PC2 (left) and PC3 (right) scores showed unimodality in all the local samples (Silverman's tests, PC2: L1, p = 0.51; L2, p = 0.83; L3, p = 0.65; L4, p = 0.91; L5, p = 0.64; L6, p = 0.94, and PC3: L1, p = 0.97; L2, p = 0.63; L3, p = 0.65; L4, p = 0.58; L5, p = 0.76; L6, p = 0.69). For sample codes, see Fig 1 and Table 1.
(DOCX)

**S2 Fig. Histograms of PC2 and PC3 of local samples of *Pseudogobio esocinus* in rivers.** Distributions of PC2 (left) and PC3 (right) scores showed unimodality in all the local samples (Silverman's tests, PC2: R1, p = 0.76; R2, p = 0.83; R3, p = 0.86; R4, p = 0.86; R5, p = 0.61; R6, p = 0.83; R7, p = 0.61, and PC3: R1, p = 0.84; R2, p = 0.79; R3, p = 0.38; R4, p = 0.94; R5, p = 0.7; R6, p = 0.33; R7, p = 0.57). For sample codes, see Fig 1 and Table 1.
(DOCX)

**S3 Fig. Boxplots of PC2, PC3, and PV*i* of local samples of *Pseudogobio esocinus*.** These boxplots show the median (dark horizontal line), 25% and 75% quartiles (the box), and the entire range (the whiskers). Distinct letters indicate significant differences in the scores in multiple comparison test under Bonferroni correction (significance level = 0.05). For sample codes, see Fig 1 and Table 1.
(DOCX)

**S4 Fig. Diet compositions of *Pseudogobio esocinus*.** Each specimen was collected from L4 (upper) and L6 (lower) in Lake Biwa.
(DOCX)

**S5 Fig. Relationship between phenotypic distances (PC2, PC3, and the ED) and genetic distance of individuals.** All the relationships were not significant (PC2, Mantel statistic r =

-0.014, P = 0.65; PC3, Mantel statistic r = -0.031, P = 0.85; ED, Mantel statistic r = -0.035, P = 0.86).
(DOCX)

## Acknowledgments

We thank Keiji Wada and all concerned members of Laboratory of Animal Ecology (Kyoto Univ.) for discussion. We also thank Eiso Inoue for advising in diet analysis, and Koji Tominaga, professional fishermen, and Lake Biwa Museum (Shiga Prefecture, Japan) for their cooperation in fish sampling and the permission to use stocking samples in the museum. We express sincere acknowledgements to Professor Donald G. Miller (California State University, Chico) and Professor Derek A. Roff (University of California, Riverside) for helpful comments and English correction.

## Author Contributions

**Conceptualization:** Chiharu Endo, Katsutoshi Watanabe.

**Data curation:** Chiharu Endo.

**Formal analysis:** Chiharu Endo.

**Funding acquisition:** Katsutoshi Watanabe.

**Investigation:** Chiharu Endo.

**Methodology:** Chiharu Endo, Katsutoshi Watanabe.

**Supervision:** Katsutoshi Watanabe.

**Writing – original draft:** Chiharu Endo, Katsutoshi Watanabe.

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
