## [Decision Letter · Decision Letter 0]

11 Feb 2020

PONE-D-19-32353

Morphological variation associated with trophic niche expansion within a lake population of a benthic fish

PLOS ONE

Dear Ms Endo,

Thank you for submitting your manuscript to PLOS ONE. After careful consideration, we feel that it has merit but does not fully meet PLOS ONE’s publication criteria as it currently stands. Therefore, we invite you to submit a revised version of the manuscript that addresses the points raised during the review process.

I only received one review of your manuscript. However, I would like to go ahead to suggest you to have a minor revision of your article based on the excellent comments provided by the reviewer. Please consider a language editing to make the article more readable. 

We would appreciate receiving your revised manuscript by Mar 27 2020 11:59PM. To enhance the reproducibility of your results, we recommend that if applicable you deposit your laboratory protocols in protocols.io, where a protocol can be assigned its own identifier (DOI) such that it can be cited independently in the future. For instructions see: http://journals.plos.org/plosone/s/submission-guidelines#loc-laboratory-protocols

We look forward to receiving your revised manuscript.

Kind regards,

Zuogang Peng, Ph.D.

Academic Editor

PLOS ONE

Journal Requirements:

2. In your Methods section, please provide additional location information of the sampling locations, including geographic coordinates for the data set if available.

4. To comply with PLOS ONE submissions requirements, please provide methods of sacrifice in the Methods section of your manuscript.

Reviewers' comments:

Reviewer's Responses to Questions

**Comments to the Author**

1. Is the manuscript technically sound, and do the data support the conclusions?

Reviewer #1: Yes

2. Has the statistical analysis been performed appropriately and rigorously? 

Reviewer #1: Yes

3. Have the authors made all data underlying the findings in their manuscript fully available?

Reviewer #1: Yes

4. Is the manuscript presented in an intelligible fashion and written in standard English?

Reviewer #1: Yes

5. Review Comments to the Author

Reviewer #1: The authors have described morphological variation in a non-model fish species associated with lentic and lotic habitats in the Lake Biwa catchment of Japan. They show divergence in mouth traits between the lake and stream populations, and link this to differences in diet from a small sub-sample of sites. Quite plausibly, they suggest the presence of more mobile prey in lakes contributes to the observed differences thereby leading to niche expansion. However, using microsatellite analysis revealed no genetic differentiation between lake and stream populations, suggesting that trait differentiation is not fixed, but instead the result of phenotypic plasticity. The necessary caveats are issued with the authors highlighting the need for experimentation to further investigate the mechanisms behind the observed patterns.

Overall, the research has been performed and reported competently. Thus, I have little in the way of major criticisms other than some suggestions for the abstract and data analysis.

1. Generally the writing is good, but I found the first few sentences of the Abstract a bit rough. Perhaps it might work better to highlight what theory predicts, before shifting to the real world. For example: “Ecological theory suggests that generalist species should have traits with multiple adaptive peaks. Consequently, in heterogeneous environments such adaptive landscapes may lead to phenotypic divergence that becomes fixed in populations via reproductive isolation, thus driving speciation. However, in wild populations ecological diversification is not always associated with obvious trait divergence and reproductive isolation in the absence of geographical barriers. Hence, the ecological conditions that promote divergence and the underlying mechanisms are not fully understood.”

2. I wondered if quantile regression might be more appropriate for testing the relationships between prey availability and morphological variation (Fig.6). I think it would be more likely that there is a “wedge-shaped” relationship, with highly mobile prey availability (i.e., proportion of amphipods in diet) forming an upper bound on morphological traits (i.e. consistent with that hypothesized at lines 491-493. See the “quantreg” package in R.

3. Fish were sampled in different years, meaning there is unaccounted variation potentially explaining some of the differences in traits. Can the authors show that temporal effects are negligible? One way to do this might be to demonstrate that habitat sampling was not temporally auto-correlated – i.e., sampling of lake and stream habitats was evenly spread across time. This could become more of an issues with the fish sampled for diets, since less sites were used. Obviously more extensive sampling with additional methods (e.g., use of biomarkers like stable isotopes for more time-integrated estimates of dietary contributions) would be also be useful here, but I appreciate the use of the dietary data to draw a line between the morphological differences and potential ecological mechanisms underpinning them.

Minor comments

1. Abstract, L33. “whole lake populations” = “the Lake Biwa catchment”

2. Abstract, L37. “the diverse habitat” = “heterogeneous environments”

3. Introduction, L40. This sentence seems a bit off – what about: “Trait variability leading to ecological niche expansion is an important factor contributing to intra- and interspecific diversity”?

4. L55. “two specialist morphs”

5. L75. “fish species”

6. L82-83. “bottom environments” = “benthic habitats”.

7. L91. “the diverse environment” = “heterogeneous environments”

8. L100. State the age of Lake Biwa here e.g., “400,000 years old”

9. L157. Components are not statistically independent – each component is part of an uncorrelated orthogonal basis set.

10. L160. How were the contributions of PC2 and PC3 calculated after removing PC1?

11. L171. What distribution was assumed in the GLMM?

12. L293-294. Does this exclude the influence of PC1, which was more associated with body size?

13. L350. “bottom environments” = “benthic habitats”

14. L425. This is controlling for body size correct?

15. L450. Wording a bit rough here. What about “P. esocinus generally forages via suction of benthic substrates and subsequent sorting of prey from fine inorganic particles. Benthic suction is achieved by a downward protrusion of the jaw whilst opening and closing the mouth.”

16. L460. I agree with this point.

17. L515. “400,000 years ago”

18. L528. “…be an important factor helping the species to increase in population size and thrive in both lake and riverine habitats.” This suggests the niche requirements of this species are now more favorable than before – perhaps due to widespread sedimentation in the Lake Biwa catchment as a result of land use change? Perhaps an interesting thought.

6. PLOS authors have the option to publish the peer review history of their article (what does this mean?). If published, this will include your full peer review and any attached files.

Reviewer #1: No

---

## [Author Response · Author response to Decision Letter 0]

26 Mar 2020

Thank you for reviewing our manuscript. We sincerely appreciate the constructive comments from you. Here we have revised our manuscript according to your comments. We hope it has adequately addressed your comments. Line numbers in this response letter are based on the revised version without tracking. Our responses are shown after the symbols, “>>”.

Editor Comments to Author:

Dear Ms Endo,

Thank you for submitting your manuscript to PLOS ONE. After careful consideration, we feel that it has merit but does not fully meet PLOS ONE’s publication criteria as it currently stands. Therefore, we invite you to submit a revised version of the manuscript that addresses the points raised during the review process.

I only received one review of your manuscript. However, I would like to go ahead to suggest you to have a minor revision of your article based on the excellent comments provided by the reviewer. Please consider a language editing to make the article more readable. 

>> We are very happy to read this comment. We have the manuscript rechecked by native speakers. 

To enhance the reproducibility of your results, we recommend that if applicable you deposit your laboratory protocols in protocols.io, where a protocol can be assigned its own identifier (DOI) such that it can be cited independently in the future.

>> We are sorry for not depositing our laboratory protocols because of absence of a formal one, but we hope that descriptions of methods and results in our manuscript would provide appropriate explanations regarding reproducibility of our experiments. 

Journal Requirements:

>> We have carefully done that in the revised version.

2. In your Methods section, please provide additional location information of the sampling locations, including geographic coordinates for the data set if available.

>> We added the information of geographic coordinates of each sampling locality in the revised version of Table 1. 

>> We described the full name of the authority and briefly explained why special permissions were not necessary for our experiments in the Ethics statement section of the revised manuscript (Lines 79-86). 

4. To comply with PLOS ONE submissions requirements, please provide methods of sacrifice in the Methods section of your manuscript.

>> We provided the methods of sacrifice in the revised manuscript (Lines 115-117). 

Review Comments to the Author: 

Reviewer #1: The authors have described morphological variation in a non-model fish species associated with lentic and lotic habitats in the Lake Biwa catchment of Japan. They show divergence in mouth traits between the lake and stream populations, and link this to differences in diet from a small sub-sample of sites. Quite plausibly, they suggest the presence of more mobile prey in lakes contributes to the observed differences thereby leading to niche expansion. However, using microsatellite analysis revealed no genetic differentiation between lake and stream populations, suggesting that trait differentiation is not fixed, but instead the result of phenotypic plasticity. The necessary caveats are issued with the authors highlighting the need for experimentation to further investigate the mechanisms behind the observed patterns. 

Overall, the research has been performed and reported competently. Thus, I have little in the way of major criticisms other than some suggestions for the abstract and data analysis.

>> Thank you for reviewing our manuscript and the helpful comments. We are very happy that you thought highly of our manuscript. Our new version of the manuscript has been revised according to your suggestions and advice as explained below. 

1. Generally the writing is good, but I found the first few sentences of the Abstract a bit rough. Perhaps it might work better to highlight what theory predicts, before shifting to the real world. For example: “Ecological theory suggests that generalist species should have traits with multiple adaptive peaks. Consequently, in heterogeneous environments such adaptive landscapes may lead to phenotypic divergence that becomes fixed in populations via reproductive isolation, thus driving speciation. However, in wild populations ecological diversification is not always associated with obvious trait divergence and reproductive isolation in the absence of geographical barriers. Hence, the ecological conditions that promote divergence and the underlying mechanisms are not fully understood.”

>> We appreciated your kind suggestion for this part. We agreed with your suggestion and changed the first few sentences in the Abstract as follows. 

Corrected sentences: Ecological theory suggests that generalist species should have traits with multiple adaptive peaks. Consequently, in heterogeneous environments such adaptive landscapes may lead to phenotypic divergence that becomes fixed in populations via reproductive isolation, thus driving speciation. However, contrary to this expectation, the process of ecological diversification in wild populations is not always associated with obvious trait divergence and reproductive isolation due to some ecological and geographic constrains. To examine the ecological conditions that promote or inhibit divergence is quite important to improve our understanding of the underlying mechanisms. (Lines 2-9)

2. I wondered if quantile regression might be more appropriate for testing the relationships between prey availability and morphological variation (Fig.6). I think it would be more likely that there is a “wedge-shaped” relationship, with highly mobile prey availability (i.e., proportion of amphipods in diet) forming an upper bound on morphological traits (i.e. consistent with that hypothesized at lines 491-493. See the “quantreg” package in R.

>> Thank you for the useful comment. We tried quantile regression models to know such detailed relationship between morphology and diet. However, the sample size in our data set was too small to adapt and achieve the precision in the process of the function fitting. Therefore, we decided to discuss the morphology-diet relationships based on the results of linear regression. 

3. Fish were sampled in different years, meaning there is unaccounted variation potentially explaining some of the differences in traits. Can the authors show that temporal effects are negligible? One way to do this might be to demonstrate that habitat sampling was not temporally auto-correlated – i.e., sampling of lake and stream habitats was evenly spread across time. This could become more of an issues with the fish sampled for diets, since less sites were used. Obviously more extensive sampling with additional methods (e.g., use of biomarkers like stable isotopes for more time-integrated estimates of dietary contributions) would be also be useful here, but I appreciate the use of the dietary data to draw a line between the morphological differences and potential ecological mechanisms underpinning them.

>> Thank you for the insightful comments. Actually, we could not exclude unaccounted variation due to temporal effects such as differences in sampling years and seasons. Our fish sampling was not ideal enough to elucidate such unexplained factors. However, in this study we only focused on the observed correlations between mouthpart morphology and diet compositions of individuals that occurred in the single site of the lake (L4) and discuss potential ecological mechanisms underlying the intra-population diversity. So, we decided not to mention the temporal effect in the manuscript. 

Minor comments

1. Abstract, L33. “whole lake populations” = “the Lake Biwa catchment”

2. Abstract, L37. “the diverse habitat” = “heterogeneous environments”

>> Thank you for your suggestion. We changed these according to your suggestion (Lines 18, 22).

3. Introduction, L40. This sentence seems a bit off – what about: “Trait variability leading to ecological niche expansion is an important factor contributing to intra- and interspecific diversity”?

>> Thank you for pointing this out. We agreed with your comment and revised this sentence as suggested (Lines 25-26). 

4. L55. “two specialist morphs”

5. L75. “fish species”

6. L82-83. “bottom environments” = “benthic habitats”.

7. L91. “the diverse environment” = “heterogeneous environments”

>> Thank you for your suggestion. We reworded these as suggested (Lines 39-40, 60, 66, 74-75).

8. L100. State the age of Lake Biwa here e.g., “400,000 years old”

>> We added the age of this lake “over 400,000 years old” to the first sentence in Study area of Methods section (Line 89). 

9. L157. Components are not statistically independent – each component is part of an uncorrelated orthogonal basis set.

>> Thank you for pointing this out. We removed the following phrase from the manuscript: “, a set of statistically independent variables” (Line 147).

10. L160. How were the contributions of PC2 and PC3 calculated after removing PC1?

>> We added more explanation for the calculation methods in the revised manuscript as follows: “We calculated the contribution ratio of each principal component in the variation excluding PC1’s contribution. The first two components (PC2 and PC3) accounted for nearly 80 percent of the variation independent of body size variation (S1 Table).” (Lines 150-153)

11. L171. What distribution was assumed in the GLMM?

>> We assumed Gaussian distribution for PC2 and PC3 scores and gamma distribution for PVi. We added more explanation for the GLMM model analysis in the manuscript (Lines 162-163, 180). 

12. L293-294. Does this exclude the influence of PC1, which was more associated with body size?

>> Yes, you are right. To make this clearer, we added the statement “Excluding the influence of body size variation (PC1),” in the first sentence of Results section (Line 283). 

13. L350. “bottom environments” = “benthic habitats”

>> We changed this according to your suggestion (Line 339).

14. L425. This is controlling for body size correct?

>> Yes, it is. We changed the first phrase of Discussion into “The morphological variation of Pseudogobio esocinus in the Lake Biwa system was primarily explained by variation in mouth size and mouth width, excluding the effect of body size.” (Lines 411-413)

15. L450. Wording a bit rough here. What about “P. esocinus generally forages via suction of benthic substrates and subsequent sorting of prey from fine inorganic particles. Benthic suction is achieved by a downward protrusion of the jaw whilst opening and closing the mouth.”

>> Thank you for your helpful suggestion. We revised the sentences as suggested (Lines 436-439). 

16. L460. I agree with this point.

>> We are happy to hear that.

17. L515. “400,000 years ago”

>> We changed this as suggested (Lines 500-501).

18. L528. “…be an important factor helping the species to increase in population size and thrive in both lake and riverine habitats.” This suggests the niche requirements of this species are now more favorable than before – perhaps due to widespread sedimentation in the Lake Biwa catchment as a result of land use change? Perhaps an interesting thought.

>> That is an interesting point. We are also interested in how and why the niche requirements of this species were changed. Further investigations centered on the effects of variations of geological and physical environments as well as the ecological and physiological conditions are likely to be fruitful. 

Addition to the above corrections, we rephrased the last sentence of Discussion as follows: “This ability may be an important factor helping the species to increase in population size and thrive in both lake and riverine habitats.” (Lines 513-515)

---

## [Editor Report · Decision Letter 1]

8 Apr 2020

Morphological variation associated with trophic niche expansion within a lake population of a benthic fish

PONE-D-19-32353R1

Dear Dr. Endo,

We are pleased to inform you that your manuscript has been judged scientifically suitable for publication and will be formally accepted for publication once it complies with all outstanding technical requirements.

With kind regards,

Zuogang Peng, Ph.D.

Academic Editor

PLOS ONE
---

## [Editor Report · Acceptance letter]

13 Apr 2020

PONE-D-19-32353R1 

Morphological variation associated with trophic niche expansion within a lake population of a benthic fish 

Dear Dr. Endo:

I am pleased to inform you that your manuscript has been deemed suitable for publication in PLOS ONE. Congratulations! Your manuscript is now with our production department. 

With kind regards,

on behalf of

Dr. Zuogang Peng 

Academic Editor

PLOS ONE